# Mechanisms of Mitochondrial Transfer Through TNTs: From Organelle Dynamics to Cellular Crosstalk

**DOI:** 10.3390/ijms262110581

**Published:** 2025-10-30

**Authors:** Margherita Zamberlan, Martina Semenzato

**Affiliations:** 1Department of Cell and Developmental Biology, Feinberg School of Medicine, Northwestern University, 303 E. Superior St., Simpson Querrey 8-522, Chicago, IL 60611, USA; margherita.zamberlan@northwestern.edu; 2Department of Biology, University of Padova, Via U. Bassi 58B, 35121 Padova, Italy; 3Veneto Institute of Molecular Medicine, Via Orus 2, 35129 Padova, Italy

**Keywords:** tunneling nanotubes, mitochondria, Miro1, mitochondrial transplantation

## Abstract

Tunneling nanotubes (TNTs) are dynamic, actin-based intercellular structures that facilitate the transfer of organelles, including mitochondria, between cells. Unlike other protrusive structures such as filopodia and cytonemes, TNTs exhibit structural heterogeneity and functional versatility, enabling both short- and long-range cargo transport. This review explores the mechanisms underlying mitochondrial transfer via TNTs, with a particular focus on cytoskeletal dynamics and the role of key regulatory proteins such as Miro1, GFAP, MICAL2PV, CD38, Connexin 43, M-Sec, thymosin β4, and Talin 2. Miro1 emerges as a central mediator of mitochondrial trafficking, linking organelle motility to cellular stress responses and tissue repair. We delve into the translational implications of TNTs-mediated mitochondrial exchange in regenerative medicine and oncology, highlighting its potential to restore bioenergetics, mitigate oxidative stress, and reprogram cellular states. Despite growing interest, critical gaps remain in understanding the molecular determinants of TNT formation, the quality and fate of transferred mitochondria, and the optimal sources for mitochondrial isolation. Addressing these questions will be essential for harnessing TNTs and mitochondrial transplantation as therapeutic tools.

## 1. Introduction

Mitochondria are essential organelles present in nearly all eukaryotic cells. They evolved through an ancient endosymbiotic process enabling complex multicellular life. Since the late 1970s, clear evidence related to mitochondria transfer has been observed from yeast protoplasts to several eukaryotic cells [1]. Beyond their well-known role in energy metabolism, these organelles also perform critical functions in cell signaling, viral sensing, calcium and redox homeostasis, iron handling, heme and lipid synthesis, cell division and cell death. We require a thorough understanding of the mechanisms underneath their transfer both in physiological and pathological conditions for their potential therapeutic readouts [2]. Recently, the International Committee on Mitochondrial Transfer and Transplantation (ICMTTN) was organized to establish the definition of basic terms, concepts and definitions [3].

Intercellular mitochondria transfer (ICMT) has been defined here as an evolutionarily conserved process where one donor cell delivers some of its mitochondria to an acceptor cell. Overall, the transfer of mitochondria can occur in contact-dependent or contact-independent mechanisms, based on the requirement of a direct cell-to-cell contact.

Under contact-independent mitochondria transfer, extracellular mitochondria can be delivered in a contact cell as free mitochondria, as extracellular vesicles (EV-Mito), which are heterogeneous in size (120 nm to 3–4 mm), and as mitochondria-derived vesicles (MDVs) containing mitochondria components, thus having a smaller size (70–150 nm). In EVs, mitochondria or mitochondrial components can be packaged into microvesicles or exosomes and released into the extracellular environment. These vesicles can then be taken up by recipient cells, contributing to bioenergetic support and signaling. MDVs are instead smaller than full mitochondria and can carry specific mitochondrial cargo to other cells or organelles, potentially influencing metabolic or immune responses [4].

From the mechanistic point of view, the most intriguing mitochondria transfer processes are the ones requiring direct contact between cells. Among them, cellular fusion cannot be classified as mitochondrial transfer, as there is no distinct donor or recipient cell involved. Thus, only adhesion-mediated transfer, dendritic structure-mediated transfer and tunneling nanotubes (TNTs) are included among them. While adhesion-mediated transfer is currently considered as the least common, mitochondria can transfer through gap junctions formed by connexin proteins (such as Connexin 43), enabling the acceptor cell to obtain mitochondria through endocytosis, rather than a direct passage through a small gap junction pore. Dendritic structure-mediated passage consists of a tubular extension (1.5–3 mm) between the donor and the acceptor cell, with a widened termini of a dendritic structure of the donor cell that is in contact with the cell membrane without fusing. This type of transfer is highly specialized, and it has been observed in osteocytes as donor [5,6] or astrocytes to neurons [7]. Finally, TNTs are actin-based cytoplasmic bridges allowing direct mitochondria movement between cells. TNTs are especially active under stress, such as hypoxia or inflammation, and have been observed in various cell types including astrocytes, mesenchymal stem cells (MSCs), and immune cells.

Generally described as dynamic, F-actin-based membranous conduits, TNTs enable long-distance intercellular communication. They were first described in rat pheochromocytoma PC12 cells [8] as an actin-based protrusion between cells enabling cytoplasmatic continuity for both the passive and active transportation of small molecules and organelles. These unique structures result of extreme interest, especially in the context of mitochondria transportation among cells. Indeed, in relevant pathological conditions where mitochondrial dysfunction contributes to cellular injury, such as neurodegeneration, cardiovascular disease, cancer, and inflammatory disorders, TNTs could be fundamental as a tool for cell recovery. For this reason, mitochondrial transplantation is emerging due to its extremely promising therapeutic potential and in this context with TNTs recognized as key mediators of this process. Nonetheless, the lack of mechanistic clarity in TNTs biogenesis, structural organization and functional regulation are still a critical barrier toward a possible clinical application.

The purpose of this review is to synthesize and critically evaluate recent advances in our understanding of mitochondrial transfer through TNTs, with particular emphasis on their biogenesis, structural characteristics, and potential applications in translational medicine.

## 2. TNTs Biogenesis and Structure

The first mitochondria transfer via TNTs was observed by electron microscopy analysis from mesenchymal stem cells to cardiomyocytes [9]. TNTs biogenesis is still not completely understood and involves multiple, sometimes overlapping mechanisms (Figure 1). The actin-driven protrusion model consists of a structure where a cell extends a long protrusion toward a donor cell. This mechanism is orchestrated by cytoskeletal regulators including M-Sec (also known as TNFaip2), which interacts with RalA GTPase and the exocyst complex to initiate membrane protrusion and actin remodeling [10]. Moreover, small Rho family GTPases—particularly Cdc42 and Rac1—modulate actin polymerization in TNTs formation, with context-dependent effects: in immune cells, Cdc42 promotes protrusion, whereas in neuronal cells, its inhibition can enhance TNT development [11]. Notably, the interplay between linear actin elongation and branched actin networks is tightly balanced: inhibition of Arp2/3 shifts polymerization toward linear filaments, facilitating TNTs extension via an Eps8–IRSp53 module [12]. Another mechanism uncovered is the cell-displacement or filopodial bridge model. This mechanism proposes the separation of two cells from one another enabling the extension of a membrane tube in which actin polymerizes. These mechanisms may act in tandem depending on the mechanical environment and the interplay of adhesion and cytoskeletal dynamics [13].

Recently another mechanism involving tip-to-tip connections has been proposed for the formation of close ended TNTs, which allows bi-directional ionic transfer between cells [14]. Briefly the TNTs develop from a double filopodia bridge created by two filopodia and by mechanical energy accumulation is leading to a twisting of the filopodia structure in un intercellular cadherin-cadherin interactions and to the formation of close ended TNTs.

It is important to distinguish TNTs from other thin cytoplasmic protrusions such as cytonemes or filopodia, which differ in morphology, mechanism of formation, and function. While TNTs are open-ended conduits enabling cytoplasmic continuity and the transfer of cellular components, cytonemes are typically closed-ended filopodia specialized for juxtacrine or paracrine signaling. Different from these other actin-based structures, TNTs are heterogeneous in their composition [15]. Structurally, TNTs exhibit remarkable heterogeneity and functional versatility. Thin TNTs, typically 20–700 nm in diameter and composed solely of actin, are highly flexible and suited for transporting small vesicles, signaling molecules, or ions. In contrast, thick TNTs—exceeding 700 nm—integrate microtubules along with actin, enabling more rigid architecture and facilitating long-range, directional transport of large cargoes such as mitochondria, but also endosomes, and lysosomes via motor proteins like kinesin, dynein, or myosin. Both types of filaments can co-exist in the same cell type, leading to the transportation of mitochondria over a distance of 150 mm [16].

## 3. Mitochondrial Mobilization Along TNTs: Miro1 Mediated Transport

Over the past decade, Miro1 (mitochondrial Rho GTPase 1, RHOT1) has been established as a critical modulator of mitochondrial transfer via TNTs, with studies across diverse tissue models linking this process to enhanced cellular and mitochondrial functional recovery. Miro1 is an outer mitochondrial membrane protein with two GTPase domains and two EF-hand Ca^2+^-binding motifs, acting as a Ca^2+^-sensitive adaptor that links mitochondria to microtubule motors via TRAK/Milton proteins to control long-range trafficking, positioning, and quality control [17,18,19]. First identified by Ahmad et al. [20] as essential for mitochondrial motility across TNTs between mesenchymal stem cells and epithelial cells, Miro1 was the earliest protein shown to accelerate intercellular mitochondrial transfer [20,21]. Subsequent studies demonstrated that Miro1 overexpression in MSCs enhances TNTs-mediated mitochondrial transfer to injured neurons, cardiomyocytes, and epithelial cells, improving outcomes in models of stroke, cardiac ischemia, and acute lung injury [22,23,24,25,26,27,28,29,30,31].

In cardiomyocytes, Miro1 drives mitochondrial transfer from human iPSC-derived MSCs; suppression of Miro1 markedly reduces transfer efficiency and diminishes cardio-protection [22].

Similarly, in the central nervous system [23], Miro1 has been shown to be a pivotal driver of intercellular mitochondrial transfer from multipotent mesenchymal stem cells (MMSCs) to neural cells, particularly under ischemic stress, where mitochondrial donation restores astrocyte bioenergetics and proliferation [24]. In a rodent stroke model, Miro1 overexpression enhanced mitochondrial delivery and significantly improved neurological recovery [24]. Boukelmoune et al. [25], showed instead that MSCs transfer mitochondria to neural stem cells (NSCs) to protect them from cisplatin-induced neurotoxicity, an effect enhanced by Miro1 overexpression. In vivo, MSC treatment preserved DCX^+^ neural progenitors in the subventricular zone and dentate gyrus, mitigating chemotherapy-induced neurogenic loss [25]. English et al. [26] described further cisplatin effects in cortical neurons and astrocytes. Co-culture experiments showed that astrocytes transfer healthy mitochondria in a Miro1 dependent manner to cisplatin-damaged neurons, thereby restoring mitochondrial potential, normalizing calcium dynamics, and improving survival [26]. Gao et al. [27] highlighted how mitochondria can be transferred bidirectionally between neural cells, a process regulated by CD38/cADPR signaling and Miro1/2, which enhances recipient cell bioenergetics [27]. Following work reconfirmed the essential role of Miro1 in mitochondrial transfer across TNTs upon stroke condition [28]. In vitro, hydrogen peroxide–damaged neurons were rescued by MSC mitochondrial donation in a Miro1-dependent manner, with overexpression enhancing and knockdown impairing neuroprotection [28].

Beyond neural and cardiac systems, Miro1-mediated mitochondrial trafficking supports other tissue types. In intervertebral disk regeneration, MSCs transfer mitochondria to nucleus pulposus cells via TNTs, rescuing cells from mitochondrial dysfunction and apoptosis [29]. In skeletal tissue, mitochondrial donation from osteolineage cells to myeloid cells depends on Miro1 and regulates osteoclast differentiation and activity, impacting skeletal homeostasis and glucocorticoid-induced osteoporosis. Loss of Miro1 impairs mitochondrial transfer, promotes myeloid differentiation toward osteoclasts, alters glutathione metabolism, and enhances osteoclast activity, contributing to bone resorption and glucocorticoid-induced osteoporosis [30]. In renal tissue, MSCs deliver mitochondria to podocytes in a Miro1- and M-Sec–dependent manner, restoring mitochondrial function and reducing apoptosis in diabetic nephropathy models [31].

Finally, Miro1 facilitates pathological mitochondrial transfer in cancer, where tumor stromal cells donate mitochondria to mtDNA-deficient cancer cells (ρ0) via TNTs, restoring respiration and promoting tumor growth. Loss of Miro1 impairs mitochondrial mobility, reduces TNTs-mediated transfer, and delays tumor formation, demonstrating its critical role in intercellular mitochondrial trafficking across both physiological and pathological contexts [32].

Across systems, the emerging picture is that Miro1 functions as a rate-limiting determinant of mitochondrial transfer efficiency. It integrates cytoskeletal transport with injury-induced signaling, ensuring that healthy mitochondria are mobilized from donor cells and delivered to stressed recipients. This makes Miro1 not only a mechanistic bridge between organelle motility and intercellular communication, but also a promising therapeutic target: one that can be upregulated to enhance tissue repair or potentially inhibited to block pathological mitochondrial exchange, such as in tumor–stromal interactions.

Recent work has extended Miro1 research into dynamic 3D culture systems and disease-specific in vivo models. In regenerative contexts, Miro1 overexpression in MSCs cultured in dynamic 3D environments enhanced TNTs formation and mitochondrial delivery to endothelial cells, accelerating wound healing and angiogenesis [33].

Mechanistic analyses clarified how Miro1 engages TRAK/Milton adaptors and the cytoskeleton to position mitochondria for TNTs entry. A major advance came from Nature Communications (2024) by Ravitch et al. [34], which resolved the MIRO1–TRAK1 complex at high resolution, revealing two non-redundant binding sites that stably tether TRAK1 to mitochondria in a Ca^2+^-independent manner. This dual-site anchoring ensures robust motor engagement, providing a molecular framework for manipulating mitochondrial transfer efficiency in both physiological and therapeutic contexts. The physical interaction that locks MIRO1 to TRAK1 onto mitochondria with high stability is a crucial process in TNTs-mediated transfer, as mitochondria must be kept tightly coupled to motors over long distances and through narrow, dynamic conduits (Figure 2).

## 4. Mitochondria Shape Can Influence Their Transfer Through TNTs

In the transferring of intact organelles, the shape of mitochondrion itself plays a central role. Mitochondrial shape is determined by the regulation of two opposing processes: mitochondrial fusion and fission. Mitofusins 1 and 2 (Mfn1 and 2) in the outer mitochondrial membrane act in concert with optic atrophy 1 (OPA1) protein in the inner mitochondrial membrane to modulate the fusion and thus the elongation of the organelle. On the opposite, dynamin related protein 1 (Drp1), recruited by its adaptor proteins Mff, Mid49/51 and Fis1 are responsible for mitochondrial fragmentation [35]. Empirical evidence demonstrates that prior to translocation into TNTs, mitochondria frequently undergo Drp1-mediated fission, a controlled fragmentation that facilitates their transport through the narrow channel of the nanotube while maintaining viability. A study related to melatonin-mediated mitochondrial transfer in HT22 cells demonstrates increased fission activity, correlating with enhanced mitochondrial mobilization via TNTs [36]. This finely tuned remodeling allows intact organelles to migrate intercellularly via TNTs, enabling the restoration of mitochondrial function in recipient cells—a feature extensively documented in various contexts including rescuing apoptotic cells and supporting intercellular homeostasis. Thus, augmented mitochondrial fragmentation mediated by Drp1 is a prerequisite for successfully entering mitochondria into the TNTs lumen.

Mitochondrial dynamics are finely tuned processes that can be modulated by multiple signaling pathways through post-translational modifications of key regulatory proteins. The recruitment of Drp1 is the first step of fission process and it is governed by various post-translational modifications, including phosphorylation, ubiquitination, S-nitrosylation, and sumoylation, which collectively determines its localization and activity [37,38]. Similarly, the Drp1 adaptor proteins Mff and Mid49 undergo regulatory modifications such as AMPK-dependent phosphorylation or sumoylation, which alter their affinity for Drp1 and thus modulate the efficiency of fission complex assembly [39]. Analogously, mitochondrial fusion is subject to comparable regulatory mechanisms. The outer membrane GTPase Mfn1 exhibits altered GTPase activity upon acetylation [40], whereas MEK/ERK-mediated phosphorylation inhibits its pro-fusion capacity [41]. Mfn2, which orchestrates both mitochondrial outer membrane fusion and endoplasmic reticulum–mitochondria tethering [42,43], represents a central hub where diverse signaling inputs converge to modulate mitochondrial and cellular homeostasis under stress conditions. Within the inner mitochondrial membrane, the fusion protein Opa1 is regulated by site-specific proteolytic cleavage mediated by the metalloproteases OMA1 and YME1L. Elevated levels of reactive oxygen species (ROS) activate OMA1-dependent cleavage or directly oxidize Opa1, leading to its functional inactivation [44,45]. Furthermore, acetylation of Opa1 has been shown to modulate its fusion activity and, consequently, the maintenance of cristae structure and mitochondrial integrity [46].

Collectively, these observations underscore that mitochondrial dynamics are not static phenomena but are subjected to different signaling pathways and second messenger (such as calcium and ROS) mediated modulations that deserves to be better understood as possible mechanisms to impact on mitochondrial size and their transfer through TNTs. On the other hand, the balance between fusion and fission events is not only correlated with the plasticity of the organelle itself but plays a role in the energy supply to the organelle in responding to stress or metabolic adaptations, other than in promoting cellular responses toward apoptosis or mitophagy, an autophagic response targeting specifically mitochondria.

## 5. Cytoskeletal Proteins as Regulators of TNTs-Mediated Mitochondrial Transfer

While TNTs have been identified across various cell types, the molecular mechanisms that drive their formation and ensure their stability remain largely uncharted, offering exciting opportunities for future investigation. Building on earlier findings that mitochondria can be transferred between cells via F-actin-based TNTs [47], Yuan et al. [48] showed that treatment with the ROCK inhibitor Y-27632 promotes TNTs formation and mitochondrial transfer in retinal pigment epithelium (RPE) cells via cytoskeletal remodeling and enhanced mitochondrial motility. Similarly, Simone et al. [49] reported that glioblastoma (GBM) cells under apoptotic stress form F-actin– and GFAP-positive TNTs containing mitochondria, with analogous structures detected in human GBM tissue. Apoptotic stimuli increased TNTs formation, GFAP-positive at the tips, and co-localizing with transferred mitochondria, suggesting that, beyond its role as a GBM marker, GFAP contributes structurally and functionally to TNT-mediated communication. In lung cancer cells, Wang et al. [50] identified MICAL2PV as an endogenous suppressor of TNTs formation and mitochondrial distribution; its downregulation promoted TNTs assembly via Miro2, enhancing mitochondrial transfer and reducing chemotherapy-induced cytotoxicity. Additional regulators have been described, including CD38, which Marlein et al. [51] proposed to be essential for TNTs formation and mitochondrial exchange in multiple myeloma, in vitro and in vivo. The gap junction protein Connexin 43 (Cx43), initially proposed by Islam et al. [52] and later confirmed by Yao et al. [53], also modulates TNTs biogenesis and mitochondrial transfer between iPSC-derived mesenchymal stem cells and epithelial cells. Barutta et al. [54] highlighted the role of the cytosolic protein M-Sec in facilitating TNTs-mediated mitochondrial transfer within glomeruli, where it rescued podocyte function in a model of focal segmental glomerulosclerosis. While M-Sec had previously been implicated in TNTs formation in macrophages and during osteoclast differentiation [55,56], this study was the first to establish its role in mitochondrial transfer. More recently, Zhang et al. [57] demonstrated that thymosin β4 (Tβ4) enhances TNTs formation between adipose-derived stem cells (ADSCs), adipocytes, and human umbilical vein endothelial cells (HUVECs) in the context of fat grafting. Tβ4 pretreatment promoted Rac/F-actin–mediated TNTs biogenesis, facilitated mitochondrial transfer to adipocytes and endothelial cells, and improved graft survival by mitigating oxidative stress and enhancing vascularization.

Baldwin et al. [58] identified Talin 2 as essential for mitochondrial transfer across TNTs between bone marrow stromal cells (BMSCs) and CD8^+^ T cells. Notably, mitochondrial transfer requires Talin 2 expression in both donor and recipient cells. CRISPR-mediated knockout of Talin 2 in either BMSCs or CD8^+^ T cells significantly impaired mitochondrial transfer, while double knockout produced an additive effect, underscoring the bidirectional requirement for Talin 2 in nanotube-mediated organelle exchange.

## 6. Tunneling Nanotubes (TNTs) and Mitochondrial Transfer in Pathological Condition

Among TNTs cargo, mitochondria have emerged as particularly significant, as their transfer can restore bioenergetics, attenuate oxidative stress, and promote survival of stressed cells. As mentioned in the previous paragraphs of this review, TNTs mediated intercellular mitochondrial transfer has been studied in different contexts including nervous, immune and cardiovascular systems, other than cancer. In the next paragraph, we will delve into the main discoveries and possible therapeutic interventions in the main pathological scenario involving TNTs and intact mitochondria.

### 6.1. Tunneling Nanotubes and Mitochondrial Transfer in Cancer: Mechanisms of Adaptation, Resistance, and Immune Modulation Across Tumor Types

Across a wide spectrum of malignancies—including glioblastoma, acute myeloid leukemia, T-cell acute lymphoblastic leukemia, hepatocellular carcinoma, breast cancer, bladder cancer, melanoma, and head and neck squamous cell carcinoma—TNTs have emerged as key mediators of tumor progression, metabolic adaptation, immune evasion, and therapy resistance [58,59,60,61,62,63,64,65,66,67]. In addition to the above cited findings, we will report here some recent examples of TNTs mitochondrial transfer across different tumor types. TNTs play a central role in GBM therapy resistance and tumor progression [59,60]. In GBM stem-like cells (GSLCs) derived from the external and infiltrative tumor zones, TNTs were shown to form in both 2D cultures and 3D tumor organoids, where they mediated functional mitochondrial transfer. Within organoids, TNTs coexisted with tumor microtubes (TMs), together establishing an interconnected tumor network [59]. Interestingly, GSLCs displayed distinct responses to irradiation in terms of TNTs induction and mitochondrial trafficking, suggesting heterogeneity in TNT-dependent adaptation. Beyond GSLC networking, TNTs also protect GBM cells from chemotherapy- and radiotherapy-induced cytotoxicity. Under oxidative stress and exposure to temozolomide (TMZ) or ionizing radiation, TNTs mediate the transfer of O6-methylguanine-DNA methyltransferase (MGMT) protein from resistant to sensitive cells, conferring resistance independently of MGMT transcription [60]. Patient tumor samples corroborate this mechanism by showing MGMT colocalization with TNT biomarkers. Furthermore, in the tumor microenvironment, astrocytes establish TNT connections with GBM cells, diminishing sensitivity not only to TMZ but also to vincristine and clomipramine. Through mitochondrial trafficking, astrocytes further enhance GBM cell survival, resistance to apoptosis, and aggressiveness in both 2D and 3D co-culture models [59,60]. In Acute Myeloid Leukemia (AML), blasts exploit TNTs to import mitochondria from BMSCs, enhancing survival and metabolic fitness; this transfer is driven by NOX2-generated superoxide and amplified by chemotherapy, potentially fueling relapse. NOX2 impairs mitochondrial transfer and selectively reduces AML viability, offering a therapeutic window that spares normal hematopoietic cells [61]. Upon chemotherapy treatment for T-cell Acute Lymphoblastic Leukemia (T-ALL), T-ALL cells export mitochondria to MSCs via TNTs in an ICAM-1–mediated manner, reducing intracellular ROS and promotes chemoresistance: blocking TNTs formation or adhesion abolishes MSC-induced protection. The directionality of mitochondrial transfer reflects distinct metabolic states: T-ALL cells favor glycolysis and offload mitochondria, while AML cells rely on oxidative phosphorylation and import them [62]. In hepatocellular carcinoma (HCC), TNTs mediate mitochondrial exchange between highly invasive and less invasive cells, thereby enhancing the migratory and invasive potential of the latter. This process is amplified under hypoxic conditions, which upregulate the mitochondrial transport protein RHOT1 (MIRO1) and the TNT formation–associated protein RAC1. High mobility group box 1 (HMGB1), a nonhistone chromatin-associated protein implicated in multiple malignancies, further promotes mitochondrial transfer by activating the NF-Y complex (via NFYA and NFYC), thereby increasing RHOT1 expression [63]. In parallel, HMGB1 enhances RAC1 aggregation at the plasma membrane under hypoxia, facilitating TNTs formation. These coordinated interactions drive mitochondrial transfer, tumor aggressiveness, and therapy resistance, consistent with clinical data showing that high HMGB1, RHOT1, or RAC1 expression correlates with shorter overall survival in HCC patients [63]. In bladder cancer, TNTs enable intercellular mitochondrial transfer from aggressive T24 cells to less invasive RT4 cells, driving phenotypic reprogramming toward greater invasiveness [64]. This exchange triggered cytoskeletal remodeling through F-actin redistribution and activated Akt/mTOR signaling and its downstream effectors, ultimately enhancing motility and invasiveness. Both in vitro and in vivo data support that TNTs-mediated mitochondrial transfer contributes to tumor heterogeneity and progression in bladder cancer. Recent work has revealed the presence of TNTs in head and neck Squamous Cell Carcinoma (SCC), both in 3D tumor spheroids and xenograft models [65]. These structures facilitate direct cargo exchange—including mitochondria, autophagosomes, and lysosomal vesicles—between distant cancer cells. A novel function of the FAK/MMP-2 signaling axis in TNT formation was uncovered, with FAK inhibition suppressing TNT assembly and MMP-2 overexpression reversing this effect. TNTs also serve as drug-delivery channels, enabling nanoparticle trafficking between epithelial and mesenchymal cancer cells. In breast cancer, the tumor microenvironment plays a pivotal role in regulating TNTs-mediated communication. Conditioned medium from infiltrating macrophages (MɸCM) enhances TNT formation in MCF-7 cells and stimulates the release of migratory cytoplasmic fragments known as microplasts [66]. These microplasts emerge through TNTs-like structures, exchange mitochondria, vesicles, and cytoplasm with parent cells, and can either reintegrate or migrate to establish new TNT-based intercellular networks, a process dependent on cytoskeletal remodeling [8]. In parallel, TNTs mediate mitochondrial transfer from adipose stem cells (ASCs) to breast cancer cells (BCCs), driving a metabolic shift toward oxidative phosphorylation (OXPHOS), elevating ATP production, and promoting chemoresistance [67]. Elevated ATP fuels ABC transporters, reducing intracellular drug retention. Notably, these effects are amplified under hypoxic conditions and can be reversed by disrupting TNT formation or mitochondrial respiration [9]. Mitochondrial transfer between cancer cells and surrounding cells in the tumor microenvironment represents a critical mechanism of immune evasion. Tumor cells can transfer mitochondria containing mutated mitochondrial DNA (mtDNA) to tumor-infiltrating lymphocytes (TILs) via TNTs and small extracellular vesicles (EVs), as reported by Ikeda et al. [62]. These shared mtDNA mutations between cancer cells and TILs are frequently observed and correlate with poor outcomes in patients receiving PD-1 blockade therapies. Transferred mitochondria disrupt mitochondrial translation, reduce oxidative phosphorylation (OXPHOS), increase glycolytic dependency, and promote T cell exhaustion, senescence, and impaired effector and memory functions. Normally, mitochondria in TILs undergo mitophagy mediated by reactive oxygen species, but co-transferred mitophagy-inhibitory molecules such as USP30 allow these mitochondria to evade clearance, leading to mitochondrial persistence and functional impairment. Importantly, blocking EV release or inhibiting USP30 activity can partially restore TIL metabolic function and effector activity, highlighting mitochondrial transfer as a potential target to enhance antitumor immunity and improve responses to immunotherapy. Conversely, therapeutic mitochondrial transfer from BMSCs to CD8^+^ T cells can enhance survival, expansion, tumor infiltration, and resistance to exhaustion across TCR, CAR, and TIL platforms [58]. Transferred mitochondria persist through cell divisions and influence epigenetic programming, providing durable functional benefits. BMSCs are ideal donors due to low HLA expression and scalable manufacturing, although transfer rates remain modest (~10%). Strategies to improve efficiency—such as identifying “super donor” subsets or targeting mitochondrial transfer machinery like TLN2—could further enhance immunotherapy outcomes.

In conclusion, across tumor types, TNTs orchestrate a complex network of intercellular communication that supports cancer survival, immune suppression, and therapy resistance. Their role in mitochondrial and organelle transfer is particularly striking, with implications for tumor progression, immune dysfunction, and therapeutic innovation. By facilitating the exchange of mitochondria and other organelles between cancer cells and the surrounding microenvironment, TNTs promote heterogeneity, enhance survival under stress, and contribute to chemoresistance, radiotherapy evasion, and metastatic potential. Importantly, mitochondrial transfer can also modulate immune responses: while cancer-derived mitochondria impair TIL function and promote immune evasion, therapeutic transfer from stromal cells such as BMSCs can enhance T cell persistence, effector function, and responsiveness to immunotherapy [58,62]. These dual roles underscore mitochondrial trafficking as both a mechanism of malignancy and a promising target for intervention. Future strategies aimed at selectively inhibiting pathogenic mitochondrial transfer while harnessing its therapeutic potential could provide novel avenues to overcome tumor resistance, improve antitumor immunity, and enhance the efficacy of current cancer treatments [68].

### 6.2. Tunneling Nanotubes and Mitochondrial Crosstalk in Cardiovascular Pathophysiology

In the context of CVDs, where mitochondrial dysfunction is a central hallmark, TNT-mediated mitochondrial transfer has become a major focus of interest for both pathophysiological understanding and potential therapeutic interventions [69,70].

Mitochondria in CVDs have been proven to play a crucial role in many cardiovascular related disorders such as ischemia–reperfusion, acute myocardial infarction, hypertrophy and cardiomyopathies [71,72]. Cardiac cells have a high energy dependency that mainly relies on fatty acid oxidation [73]. Other than this, calcium handling is a key component in the contractility of the heart, together with its high response to oxidative damage from mitochondrial derived reactive oxygen species.

In the physiological context, TNTs are crucial in the development of the heart. In neonatal cardiomyocytes, TNTs are fundamental during heart development for the transfer of mitochondria and vesicles between cardiomyocytes or between cardiomyocytes and macrophages [74,75]. An existing paper recently published in Science, finally points out the crucial roles of TNTs in the communication between endocardium and heart muscle cells during cardiac development [76].

TNTs has also been studied as a new mechanism of communication between cardiomyocytes and cardiac fibroblasts both in the co-culture system and in adult mouse hearts [77]. Using an in vitro model of ischemia–reperfusion injury, Han et al. demonstrated the beneficial effect on mitochondrial transfer from MSCs toward H9C2 cardiomyocyte [78]. In the same model, the passage of MSCs derived mitochondria into endothelial cells, a key component of cardiac tissue, had beneficial effects [79]. Vallabhaneni and colleagues demonstrate the mitochondrial transfer from vascular smooth cells to MSCs to regulate their proliferative response, rather than their differentiation [80]. In a model of cardiac hypertrophy induced by isoproterenol stimulation, bone marrow derived MSCs could rescue the phenotype, by Cx43 TNTs formation [81].

These investigations point out TNT-mediated mitochondrial transfer as a vital adaptive mechanism in the cardiovascular system, particularly under stress conditions where mitochondrial dysfunction drives pathology. By elucidating the molecular machinery and therapeutic potential of TNTs, novel strategies can be designed to enhance intercellular mitochondrial rescue, offering hope for improved interventions in ischemic and degenerative cardiovascular diseases. Mitochondrial transfer through TNTs is currently considered as a possible strategy to overcome cardiovascular disease [82].

### 6.3. Tunneling Nanotube-Mediated Mitochondrial Transfer in Neurodegenerative Diseases: A Double-Edged Sword

The role of mitochondria in intercellular communication through tunneling nanotubes (TNTs) has been extensively documented in recent comprehensive reviews [83,84,85]. In this chapter, we focus on the specific impact of TNT-mediated mitochondrial transfer in the context of neurodegenerative diseases. TNT formation contributes significantly to brain development in vivo, particularly through the regulation of neural stem and progenitor cells. During neurogenesis, mitochondria are central to metabolic reprogramming, facilitating the transition from oxidative phosphorylation to glycolysis [86,87,88,89]. Direct evidence of mitochondrial transfer via TNTs has been provided by Wang and colleagues, who demonstrated mitochondrial exchange between neural stem cells and brain endothelial cells [90].

Beyond energy metabolism, mitochondrial dysfunction contributes to multiple pathological processes in neurons, including oxidative stress, impaired calcium regulation, and disruption of intracellular signaling pathways. For example, in Drosophila melanogaster, mitochondrial impairment has been linked to dementia and amyotrophic lateral sclerosis [91]. Notably, TNT-mediated mitochondrial transfer can serve as a neuroprotective mechanism. Following ischemic stroke, pericytes donate mitochondria to astrocytes via TNTs, enhancing survival and promoting blood–brain barrier repair [92]. Similarly, cocultures of microglia and neurons have revealed TNT-dependent mitochondrial rescue, which reduces toxic protein accumulation [93].

Conversely, TNTs can also be hijacked by pathological processes. In malignancies such as glioblastoma and bladder tumors, TNTs facilitate mitochondrial transfer that sustains cancer proliferation, prevents hypoxia-induced apoptosis, and mediates therapy resistance (see Section 6.1). Comparable mechanisms appear in neurodegenerative conditions, including Parkinson’s disease (PD), Alzheimer’s disease (AD), and Huntington’s disease (HD), where toxic protein aggregates—such as α-synuclein and tau—accumulate in neurons. Scheiblich et al. (2024) demonstrated that microglia counteract this process by transferring healthy mitochondria to damaged neurons, thereby rescuing them from oxidative stress and mitochondrial dysfunction [93]. The therapeutic potential of mitochondrial supplementation has also been demonstrated in PD models through direct injection of isolated mitochondria [94,95].

At the same time, TNTs mediate the spread of pathogenic protein aggregates. Several studies have reported the transfer of tau and α-synuclein [96,97,98,99,100], as well as mutant huntingtin [101,102]. This bidirectional trafficking underscores the paradoxical role of TNTs in both neuronal repair and disease propagation.

The duality of TNT function positions them as a “double-edged sword” in therapeutic strategies: blocking TNTs could limit pathogenic spread but may also impair beneficial mitochondrial transfer. Therefore, the key challenge in neurodegenerative diseases is to design interventions that selectively enhance protective mitochondrial trafficking while simultaneously restricting harmful exchanges.

## 7. Translational Implication of Mitochondrial Transfer Across TNTs

Mitochondrial transfer via TNTs represents a paradigm-shifting mechanism with significant translational potential in regenerative medicine and oncology. By restoring bioenergetic capacity, attenuating oxidative stress, and reprogramming metabolic and epigenetic states, TNTs-mediated exchange can rescue dysfunctional cells and promote tissue repair in settings of injury, inflammation, or degeneration. Despite its therapeutic promise, the molecular mechanisms governing mitochondrial transfer across TNTs remain incompletely understood. Notably, Lin et al. [103] introduced the concept that TNTs-mediated mitochondrial transfer can exert beneficial effects on recipient cells irrespective of the metabolic status of the donor mitochondria. Specifically, they showed that MSCs transfer mitochondria to ECs via TNTs following ECs transplantation, and that this process is crucial for successful ECs engraftment in vivo [103]. A central finding was that transferred mitochondria trigger mitophagy in recipient cells through activation of the PINK1/PARKIN pathway, providing functional benefits independent of donor mitochondrial quality or quantity. Importantly, similar outcomes were observed when mitochondria were delivered by direct transplantation, indicating that even though uptake occurs via distinct mechanisms (TNTs vs. endocytosis), part of the mitochondrial pool still will induce mitophagy initiation. Notably, the study showed that mitochondrial functionality and dosage are not prerequisites for therapeutic benefit; however, in scenarios where endogenous mitochondria are impaired, exogenous mitochondria may act through alternative mechanisms to support cellular recovery.

These findings underscore the therapeutic relevance of mitochondrial transplantation, a strategy explored for treating conditions ranging from neurodegeneration to cardiovascular, liver and renal diseases [104,105,106,107,108]. Despite promising outcomes in preclinical models, several critical questions remain unanswered. The physiological status of transplanted mitochondria—whether metabolically active, damaged, or undergoing stress—is often undefined, yet may profoundly influence their integration and function in recipient cells. Moreover, the optimal source for mitochondrial isolation is still under debate. While MSCs are frequently used due to their immunomodulatory properties and mitochondrial resilience, other cell types may offer distinct advantages depending on the therapeutic context. Additionally, the downstream effects of exogenous mitochondria on recipient cell fate, signaling pathways, and long-term tissue remodeling remain poorly characterized. Addressing these gaps is essential for refining mitochondrial transplantation protocols and unlocking their full clinical potential.

### Evidence and Challenges in Demonstrating TNTs In Vivo

While TNT-mediated mitochondrial transfer has been extensively characterized in vitro, its unequivocal demonstration in vivo remains technically challenging. Most studies rely on ex vivo co-cultures, organoid systems, or indirect observations in animal models. Nevertheless, increasing evidence supports the existence of TNT-like structures and their role in organelle transfer within living tissues. In several models, manipulation of TNT-associated proteins has produced in vivo phenotypes consistent with TNT-mediated mitochondrial trafficking, even when direct imaging of the tubes was not feasible. A notable example is provided by Lin et al. [103], who demonstrated mitochondrial transfer from MSCs to ECs during cell engraftment. Histological examination of 24 h grafts revealed labeled mitochondria within TNT-like protrusions, and genetic ablation of TNFAIP2 or Miro1 in MSCs markedly reduced both TNTs formation and mitochondrial transfer, confirming a functional role for TNTs in vivo. Other studies have similarly reported donor-specific mitochondrial markers or mtDNA appearing in recipient cells after transplantation, lending indirect support to the physiological relevance of TNT-mediated transfer [51,64].

Together, these findings suggest that TNTs operate alongside other contact-dependent and vesicular mechanisms to enable mitochondrial exchange in complex tissue environments. Improved intravital imaging, correlative light–electron microscopy, and genetic labeling strategies will be essential to definitively map TNTs networks and quantify their contribution to mitochondrial transfer during intercellular communication.

## 8. Conclusions

Mitochondrial transfer via TNTs represents a transformative mechanism of intercellular communication with broad therapeutic relevance. The integration of cytoskeletal remodeling, organelle trafficking, and stress signaling enables TNTs to deliver functional mitochondria to compromised cells, promoting recovery across diverse pathological contexts. Key established regulators such as Miro1, or more recently discovered such as GFAP and Talin 2, orchestrate this process, while emerging evidence supports the feasibility of mitochondrial transplantation as a complementary strategy. However, the field faces unresolved challenges: the metabolic status of transferred mitochondria, their long-term impact on recipient cells, and the identification of optimal donor sources remain poorly defined. Future research must clarify these mechanisms to optimize TNT-mediated therapies and unlock their full clinical potential in tissue regeneration and disease modulation.

## Figures and Tables

**Figure 1 ijms-26-10581-f001:**
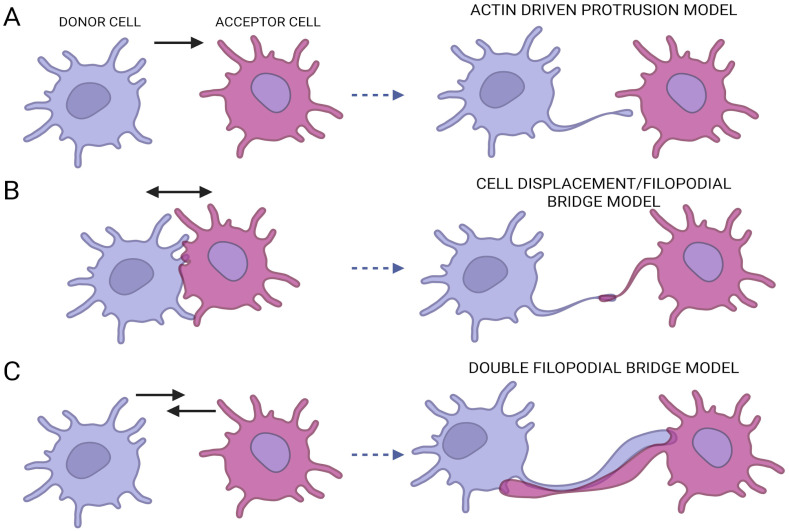
TNTs biogenesis mechanisms. (**A**) In the actin drive protrusion model, the donor cell drives a protrusion toward the acceptor cell. (**B**) Cell displacement model of single filopodial bridge model consists of two cells that separate enabling the formation of a connection between them. (**C**) Double cell filopodia consists of the formation of a double tubular structure that allows a bidirectional transfer.

**Figure 2 ijms-26-10581-f002:**
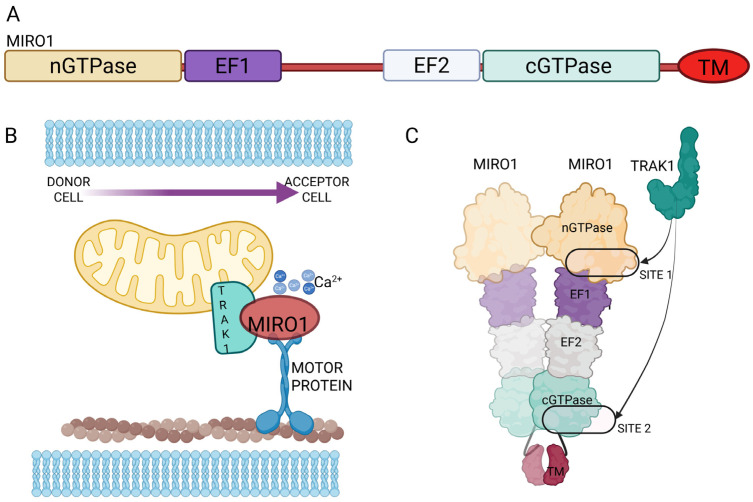
MIRO1 structure and transport: (**A**) MIRO1 protein domains containing two GTPase domains (nGTPase and cGTPase) and two EF binding domains (1 and 2). (**B**) Cartoon depicting a general interaction model between MITO1-TRAK1 and motor proteins inside TNTs. (**C**) More recent model (proposed in [34]) related to MIRO1 dimerization and the presence of two sites of interaction of TRAK1.

## Data Availability

No new data were created or analyzed in this study. Data sharing is not applicable to this article.

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
