# Peer review of "Mechanisms of Mitochondrial Transfer Through TNTs: From Organelle Dynamics to Cellular Crosstalk"

_ijms, 2025, doi:10.3390/ijms262110581_

Round 1

Reviewer 1 Report

Comments and Suggestions for Authors

I must give a credit to this work—it's very good and will undoubtedly be in high demand.

However, there are a number of issues that the authors should address.

  • The main question is to what extent the emergence of nanotubes as a means of communication between cells reflects the actual state of intercellular interactions in living tissue, rather than being an artificial process observed only during cell culture. Since we are interested in this question not only from a fundamental perspective but also from a practical aspect—namely, whether intercellular transfer can be activated in tissue by generating nanotubes—it would be necessary to more clearly describe the limitations of such studies. To my knowledge, the process of mitochondrial transfer in tissue has not been observed very often, and the work (doi: 10.5966/sctm.2015-0010) represents a very rare example of this type of observation. Although even in this case, it is unknown to what extent nanotubes were involved in the transfer of mitochondria from MSCs to neighboring cells rather than the transfer of mitochondria from MSCs to adjacent cells used gap junctions, although it is clear that cell fusion does not occur, as only single transferred mitochondria were visible.
  • The second question concerns the extent to which thin cytoplasmic filaments are similar in nature, interaction patterns, and formation mechanisms. Specifically, the authors write about the general organization of nanotubes, which begin with an expansion at the site of contact with the donor cell and end with a similar funnel-shaped expansion upon contact with the recipient. However, cytonemes observed, for example, in neutrophils, do not have such expansions, and the proposed mechanism involves actin depolymerization, rather than polymerization (e.g., see https://doi.org/10.4161/cam.23130). The question is how aware the authors are of such data and views, which requires us to ask what structures are being discussed and how to make a definition of nanotubes. Can all such cytoplasmic protrusions participate in the transfer of cellular fragments?
  • It would be good to evaluate Miro's stoichiometry for mitochondrial length. It's clear that long mitochondria aren't transported, but of what length should the minimum fragment be, and how many miro molecules are involved in this fragment—one or several, working in concert? Is it possible to obtain such estimates?

By the way, I know nothing about cytomenes (line 123). I know about cytonemes?

Please check the authors’ names in ref. 49

Author Response

We sincerely thank the reviewer for her/his encouraging and positive evaluation of our manuscript. We greatly appreciate the recognition of the relevance and potential impact of our work. We have carefully considered all the issues raised and have addressed each point in detail below.

I must give a credit to this work—it's very good and will undoubtedly be in high demand.

However, there are a number of issues that the authors should address.

  • The main question is to what extent the emergence of nanotubes as a means of communication between cells reflects the actual state of intercellular interactions in living tissue, rather than being an artificial process observed only during cell culture. Since we are interested in this question not only from a fundamental perspective but also from a practical aspect—namely, whether intercellular transfer can be activated in tissue by generating nanotubes—it would be necessary to more clearly describe the limitations of such studies. To my knowledge, the process of mitochondrial transfer in tissue has not been observed very often, and the work (doi: 10.5966/sctm.2015-0010) represents a very rare example of this type of observation. Although even in this case, it is unknown to what extent nanotubes were involved in the transfer of mitochondria from MSCs to neighboring cells rather than the transfer of mitochondria from MSCs to adjacent cells used gap junctions, although it is clear that cell fusion does not occur, as only single transferred mitochondria were visible

We thank the reviewer for this insightful comment. Although the unequivocal visualization of TNTs in vivo remains technically challenging, several studies have provided compelling ex vivo and indirect in vivo evidence supporting their existence and functional role in mitochondrial transfer. In many cases, investigators first delineated the underlying mechanisms in controlled in vitro systems and subsequently validated their hypotheses in animal models, where modulation of TNT-related pathways produced the expected physiological outcomes, even when direct imaging of TNTs was not feasible. This provides indirect but robust support for TNT-mediated mitochondrial transfer occurring in vivo (e.g. https://doi.org/10.18632/oncotarget.14695; https://doi.org/10.1158/0008-5472.CAN-18-0773). For instance, Lin et al. (Nature, 2024; https://doi.org/10.1038/s41586-024-07340-0) demonstrated mitochondrial transfer from mesenchymal stromal cells (MSCs) to endothelial cells during cell engraftment. Histological examination of 24-h grafts revealed labeled mitochondria within TNT-like structures (Fig. 1A). Furthermore, genetic ablation of TNFAIP2 or Miro1 in MSCs markedly reduced both mitochondrial transfer and TNT formation within grafts, thereby functionally validating the contribution of TNTs to in vivo mitochondrial exchange. It is also important to note that TNTs are rarely the exclusive mechanism of intercellular mitochondrial transfer—other contact-dependent or vesicular routes may coexist within the same microenvironment. To address this point in the manuscript, we have added a dedicated paragraph 7.1 (line 549-570) in the review discussing the current evidence, challenges, and recent advances supporting TNTs formation and mitochondrial transfer in vivo, with specific reference to the study by Lin et al. and related findings

  • The second question concerns the extent to which thin cytoplasmic filaments are similar in nature, interaction patterns, and formation mechanisms. Specifically, the authors write about the general organization of nanotubes, which begin with an expansion at the site of contact with the donor cell and end with a similar funnel-shaped expansion upon contact with the recipient. However, cytonemes observed, for example, in neutrophils, do not have such expansions, and the proposed mechanism involves actin depolymerization, rather than polymerization (e.g., see https://doi.org/10.4161/cam.23130). The question is how aware the authors are of such data and views, which requires us to ask what structures are being discussed and how to make a definition of nanotubes. Can all such cytoplasmic protrusions participate in the transfer of cellular fragments?

We thank the reviewer for this insightful and important question highlighting the structural and mechanistic diversity among thin cytoplasmic protrusions such as tunneling nanotubes (TNTs), cytonemes, and other actin-based extensions. We fully agree that these structures, while morphologically related, differ in their biogenesis, composition, and functional specialization. In our manuscript, the term tunneling nanotubes (TNTs) refers specifically to open-ended, F-actin–rich conduits that establish direct cytoplasmic continuity between donor and recipient cells, thereby enabling intercellular transfer of organelles and cytoplasmic components, including mitochondria. TNTs typically display a funnel-like widening at one or both termini, as described in mesenchymal and epithelial models of mitochondrial transfer (e.g., Ahmad et al., 2014; Rustom et al., 2004; Gerdes et al., 2013). By contrast, cytonemes—as observed in neutrophils and other immune cells—are primarily signaling filopodia that remain closed-ended and are not known to support cytoplasmic continuity. As the reviewer notes, their dynamics may involve actin depolymerization rather than polymerization (e.g., Galkina et al., 2012, https://doi.org/10.4161/cam.23130), underscoring mechanistic distinctions from TNTs.

To clarify this in the revised version, we have added the following text to the paragraph 2 (line 128-132):

It is important to distinguish TNTs from other thin cytoplasmic protrusions such as cytonemes or filopodia, which differ in morphology, mechanism of formation, and function. While TNTs are open-ended conduits enabling cytoplasmic continuity and the transfer of cellular components, cytonemes are typically closed-ended filopodia specialized for juxtacrine or paracrine signaling”.

In this review we wanted to highlight the importance on TNT  in mitochondria transportation, thus focusing mainly in this structures. To our knowledge, not all cytoplasmic protrusions participate in the transfer of cellular fragments. The defining feature of TNTs, as used in our study, is their ability to mediate direct intercellular exchange of organelles, rather than mere contact or signaling.

  • It would be good to evaluate Miro's stoichiometry for mitochondrial length. It's clear that long mitochondria aren't transported, but of what length should the minimum fragment be, and how many miro molecules are involved in this fragment—one or several, working in concert? Is it possible to obtain such estimates?

The question raised by the reviewer is extremely interesting. However, such estimation would be only speculative, since tp our knowledge no one ever tried to calculate Miro stoichiometry for mitochondrial movement.

By the way, I know nothing about cytomenes (line 123). I know about cytonemes?

We apologize for the mistake and correct the spelling.

Please check the authors’ names in ref. 49

We thank the reviewer for the checking. References were added using EndNote, but we manage to correct authors’ names.

Reviewer 2 Report

Comments and Suggestions for Authors

In this review, Zamberlan and Semenzato present a synthesis and critical evaluation of recent advances in our understanding of mitochondrial transfer through tunnelling nanotubes (TNTs). They focus particularly on TNT biogenesis, structural characteristics and potential applications in translational medicine. The authors clarify how TNTs deliver functional mitochondria to compromised cells, thereby promoting recovery in various pathological contexts, including cancer, cardiovascular disease, and degenerative diseases. They also specify the role of mitochondria in these conditions. They also specify the role of mitochondria in these conditions. However, they are necessary further studies to better understand the effects of different signaling pathways on mitochondrial dynamics and/or to identify additional signaling modalities that regulate mitochondrial transfer for therapeutic purposes.

Overall, the authors have addressed a very interesting research topic, and the manuscript is clearly structured and comprehensively written. The bibliography on the topic is very recent and adequate for the purpose. The illustration is clear and well executed.

However, there are minor revisions to be made.

Lines 5 and 6: Following the instructions in the journal, the information in lines 8 and 10 should be entered in lines 5 and 6 respectively, followed by the corresponding email addresses.

Line 70: Only keep the acronym 'TNTs' because it has already been mentioned in line 62.

Line 140: Include this article, which explains how mitochondria move in TNTs. doi: 10.3390/cancers13225812.

Line182: The number 31 and the appropriate square brackets are missing.

Paragraph 3, line 207: It would be appropriate to include an illustration showing the associations between mitochondria and the various proteins that determine their movement in TNTs in paragraph 3. This would help readers to understand the topic better.

Line 289: Only keep the acronym 'GBM' because it has already been mentioned in line 240.

Line 291: Replace NTs with TNTs.

Line 346: Add square brackets to the number 18.

Line 441: The author's name and parentheses should be removed from the reference.

Line 459: Remove the round bracket

Errors in spacing seem to be present in lines 215, 308, 411 and 459

Author Response

We sincerely thank the reviewer for their positive and insightful assessment of our work.

In this review, Zamberlan and Semenzato present a synthesis and critical evaluation of recent advances in our understanding of mitochondrial transfer through tunnelling nanotubes (TNTs). They focus particularly on TNT biogenesis, structural characteristics and potential applications in translational medicine. The authors clarify how TNTs deliver functional mitochondria to compromised cells, thereby promoting recovery in various pathological contexts, including cancer, cardiovascular disease, and degenerative diseases. They also specify the role of mitochondria in these conditions. They also specify the role of mitochondria in these conditions. However, they are necessary further studies to better understand the effects of different signaling pathways on mitochondrial dynamics and/or to identify additional signaling modalities that regulate mitochondrial transfer for therapeutic purposes.

As suggested by the reviewer, we agreed to add a small paragraph in which we briefly refer to some possible pathways and post translational modifications that impact on mitochondrial dynamics in paragraph 4 (line 244-268).

Overall, the authors have addressed a very interesting research topic, and the manuscript is clearly structured and comprehensively written. The bibliography on the topic is very recent and adequate for the purpose. The illustration is clear and well executed.

However, there are minor revisions to be made.

Lines 5 and 6: Following the instructions in the journal, the information in lines 8 and 10 should be entered in lines 5 and 6 respectively, followed by the corresponding email addresses.

We thank the reviewer for the suggestions. We kept following the format related to the journal. Any correction related to the position of the affiliations or addresses has been solved directly with the editorial office.

Line 140: Include this article, which explains how mitochondria move in TNTs. doi: 10.3390/cancers13225812.

We preferred to include the suggested reference in the paragraph related to TNTs as a possible target for cancer treatment (see number 68).

Line182: The number 31 and the appropriate square brackets are missing.

We corrected properly the number of the references referred to that paragraph.

Paragraph 3, line 207: It would be appropriate to include an illustration showing the associations between mitochondria and the various proteins that determine their movement in TNTs in paragraph 3. This would help readers to understand the topic better.

As suggested by the reviewer we added an additional representative figure related to paragraph 3.

Line 70: Only keep the acronym 'TNTs' because it has already been mentioned in line 62.

Line 289: Only keep the acronym 'GBM' because it has already been mentioned in line 240.

Line 291: Replace NTs with TNTs.

Line 346: Add square brackets to the number 18.

Line 441: The author's name and parentheses should be removed from the reference.

Line 459: Remove the round bracket

Errors in spacing seem to be present in lines 215, 308, 411 and 459

We thank the reviewer for the careful reading. We did all the corrections requested.

Round 2

Reviewer 1 Report

Comments and Suggestions for Authors

I think that the authors properly addressed the reviewer's critique and it is ready for publication